# NEURAL DYNAMICAL SYSTEMS: BALANCING STRUCTURE AND FLEXIBILITY IN PHYSICAL PREDICTION

## ABSTRACT

We introduce Neural Dynamical Systems (NDS), a method of learning dynamical models in various gray-box settings which incorporates prior knowledge in the form of systems of ordinary differential equations. NDS uses neural networks to estimate free parameters of the system, predicts residual terms, and numerically integrates over time to predict future states. A key insight is that many real dynamical systems of interest are hard to model because the dynamics may vary across rollouts. We mitigate this problem by taking a trajectory of prior states as the input to NDS and train it to dynamically estimate system parameters using the preceding trajectory. We find that NDS learns dynamics with higher accuracy and fewer samples than a variety of deep learning methods that do not incorporate the prior knowledge and methods from the system identification literature which do. We demonstrate these advantages first on synthetic dynamical systems and then on real data captured from deuterium shots from a nuclear fusion reactor. Finally, we demonstrate that these benefits can be utilized for control in small-scale experiments.

## 1 INTRODUCTION

The use of function approximators for dynamical system modeling has become increasingly common. This has proven quite effective when a substantial amount of real data is available relative to the complexity of the model being learned (Chua et al., 2018; Janner et al., 2019; Chen et al., 1990). These learned models are used for downstream applications such as model-based reinforcement learning (Nagabandi et al., 2017; Ross & Bagnell, 2012) or model-predictive control (MPC) (Wang & Ba, 2019).

Model-based control techniques are exciting as we may be able to solve new classes of problems with improved controllers. Problems like dextrous robotic manipulation (Nagabandi et al., 2019), game-playing (Schrittwieser et al., 2019), and nuclear fusion are increasingly being approached using model-based reinforcement learning techniques. However, learning a dynamics model using, for example, a deep neural network can require large amounts of data. This is especially problematic when trying to optimize real physical systems, where data collection can be expensive. As an alternative to data-hungry machine learning methods, there is also a long history of fitting models to a system using techniques from system identification, some of which include prior knowledge about the system drawn from human understanding (Nelles, 2013; Ljung et al., 2009; Sohlberg & Jacobsen, 2008). These models, especially in the gray-box setting, are typically data-efficient and often contain interpretable model parameters. However, they are not well suited for the situation where the given prior knowledge is approximate or incomplete in nature. They also do not generally adapt to the situation where trajectories are drawn from a variety of parameter settings at test time. This is an especially crucial point as many systems of interest exhibit path-dependent dynamics, which we aim to recover on the fly.

In total, system identification methods are sample efficient but inflexible given changing parameter settings and incomplete or approximate knowledge. Conversely, deep learning methods are more flexible at the cost of many more samples. In this paper, we aim to solve both of these problems by biasing the model class towards our physical model of dynamics. Physical models of dynamics are often given in the form of systems of ordinary differential equations (ODEs), which are ubiquitous and may have free parameters that specialize them to a given physical system. We develop a model that uses neural networks to predict the free parameters of an ODE system from the previous

timesteps as well as residual terms added to each component of the system. To train this model, we integrate over the ODE and backpropagate gradients from the prediction error. This particular combination of prior knowledge and deep learning components is effective in quickly learning the dynamics and allows us to adjust system behavior in response to a wide variety of dynamic parameter settings. Even when the dynamical system is partially understood and only a subset of the ODEs are known, we find that our method still enjoys these benefits. We apply our algorithm to learning models in three synthetic settings: a generic model of ballistics, the Lorenz system (Lorenz, 1963), and a generalized cartpole problem, which we use for control as well. We also learn a high-level model of plasma dynamics for a fusion tokamak from real data.

The contributions of this paper are

- We introduce Neural Dynamical Systems (NDS), a new class of model for learning dynamics that can incorporate prior knowledge about the system.

- We show that these models naturally handle the issue of partial or approximate prior knowledge, irregularly spaced data, and system dynamics that change across instantiations, which generalizes the typical system identification setting. We also show that these advantages extend to control settings.

- We demonstrate this model's effectiveness on a real dynamics problem relevant to nuclear fusion and on synthetic problems where we can compare against a ground truth model.

## 2 RELATED WORK

**System Identification and Deep Learning with Structure**    There is a long tradition of forecasting physical dynamics with either machine learning or techniques based on domain knowledge of the dynamics, especially in the field of system identification, where Ljung (2010), Schoukens & Ljung (2019) and Cressie & Wikle (2015) are good summaries. Often, this space is discussed as a spectrum from a purely prior-knowledge-based system (white-box) to a purely data-driven system (black-box) with several shades of gray in between.

White-box models use prior knowledge to precisely give the relationship between quantities of interest over time and there is extensive literature on solving them (Brenan et al., 1995). 'Shades of gray' may distinguish between levels of prior knowledge or how equations cover subsets of the state space (Ljung, 2010). Other prior work focuses on online parameter estimation (Vahidi et al., 2005), but this relies on an ongoing trajectory through the system and is difficult to use in our setting.

In nonlinear black-box settings, there are a variety of techniques used to solve system identification models. Volterra series, a generalization of Taylor series which respects dependency on the past, have been used for system identification (Rugh, 1981). Block models such as the Hammerstein (1930) and Weiner (Billings, 1980) models and their combination can also identify systems. Feedforward and recurrent neural networks have been widely used to model dynamical systems (Chua et al., 2018; Nagabandi et al., 2017; Hochreiter & Schmidhuber, 1997), with additional constraints on stability (Manek & Kolter, 2020) or the Hamiltonian (Chen et al., 2019) and many others added. Nonlinear autoregressive moving average models with exogenous variables (NARMAX) have also been used widely to model dynamical systems and this class is a superset of nearly everything else discussed (Brunton et al., 2015; Rahrooh & Shepard, 2009). Broadly, none of these algorithms are well-suited to a setting where the dynamic parameters of the system change across rollouts.

There have also been several approaches for including physical structure in deep models. Raissi et al. (2019) use automatic partial derivative computation to force a neural network to fit a given ODE or PDE solution. de Avila Belbute-Peres et al. (2018) uses a linear complementarity problem to differentiate through 2d physics simulations however their method is not general to more dimensions or other types of problems besides mechanics. Cranmer et al. (2019) uses graph networks to discover physical laws. Chen et al. (2019), Sanchez-Gonzalez et al. (2019) and Cranmer et al. (2020) force the network to respect Hamiltonian and Lagrangian constraints but without specific problem data on the system. Psichogios & Ungar (1992) predicts physical parameters for a given ODE model and Rico-Martinez et al. (1994) predict residuals. Thompson & Kramer (1994) similarly builds a hybrid parameter prediction function into a dynamical model. These last three works are especially similar to ours, though they use tiny networks, are problem-specific in their setup, and don't take advantage of backpropagation through a numerical ODE solver.

**Neural Ordinary Differential Equations**   As most numerical ODE solvers are algorithms involving differentiable operations, it has always been possible in principle to backpropagate through the steps of these solvers dating back to at least Runge (1895). However, since each step of the solver involves calling the derivative function, naïve backpropagation incurs an $O(n)$ memory cost, where $n$ is the number of derivative calls made by the solver. Historically Pontryagin (2018) and recently Chen et al. (2018) demonstrated that by computing gradients through the adjoint sensitivity method, the memory complexity of backpropagating through a family of ODE solvers can be reduced to $O(1)$ for a fixed network, as opposed to the naive $O(n)$. However, this work only used generic neural networks as the derivative function and did not consider dynamics modeling. They also provide a PyTorch package which we have built off of in our work.

There has been some work using neural ordinary differential equations to solve physical problems. Portwood et al. (2019) used a fully-connected neural ODE with an RNN encoder and decoder to model Navier-Stokes problems. Rudy et al. (2019) used a neural network integrated with a Runge-Kutta method for noise reduction and irregularly sampled data. There has also been work learning the structure of dynamical systems, first with a convolutional warping scheme inspired by advection-diffusion PDEs (de Bezenac et al., 2018), then with a Neural ODE which was forced to respect boundary conditions and a partial observation mechanism (Ayed et al., 2019).

**Machine Learning for Nuclear Fusion**   As far back as 1995, van Milligen et al. (1995) showed that by approximating the differential operator with a (single-layer, in their case) neural network, one could fit simple cases of the Grad-Shafranov equation for magnetohydrodynamic equilibria. Recent work has shown that plasma dynamics are amenable to neural network prediction. In particular, Kates-Harbeck et al. (2019) used a convolutional and LSTM-based architecture to predict possible plasma disruptions (when a plasma instability grows large and causes a loss of plasma containment and pressure).

There has also been work in the field of plasma control: a neural network model of the neutral beam injection for the DIII-D tokamak has been deployed in order to diagnose the effect of controls on shots conducted at the reactor (Boyer et al., 2019b). Additionally, (Boyer et al., 2019a) used classic control techniques and a simpler model of the dynamics to develop a controller that allows characteristics of the tokamak plasma to be held at desired levels. Others have used contextual Bayesian optimization to choose single-state controls which direct the plasma to desirable states (Char et al., 2019; Chung et al., 2020).

## 3   PROBLEM SETTING

Typically, a dynamical system $\dot{x} = f_\phi(x, u, t)$ with some parameters $\phi$ is the conventional model for system identification problems. Here, state is $x \in \mathcal{X}$, control is $u \in \mathcal{U}$, and time is $t \in \mathbb{R}$. The objective is to predict future states given past states, past and future controls, and prior knowledge of the form of $f$. We denote $x(\phi, t, \mathbf{u}, x_0) = x_0 + \int_0^t f_\phi(x, u, t)dt$ as the state obtained by integrating our dynamical system around $f$ to time $t$.

We consider in this work a more general setting and address the problem of prediction and control over a *class of dynamical systems*, which we define as the set $\{\dot{x} = f_\phi(x, u, t) \mid \phi \in \Phi\}$, where $\Phi$ is the space of parameters for the dynamical system (e.g. spring constant or terminal velocity). We can generate a trajectory from a class by sampling a $\phi \sim P(\Phi)$ for some distribution $P$ and choosing initial conditions and controls. In real data, we can view nature as choosing, but not disclosing, $\phi$. For a particular example $j$, we sample $\phi \sim P(\Phi)$ and $x_0 \sim P(X_0)$ and are given controls $\mathbf{u}$ indexed as $u(t)$ and input data $\{x(\phi, t_i, \mathbf{u}, x_0)\}_{i=0}^T$ during training. At test time, we give a shorter, prefix time series $\{x(\phi, t_i, \mathbf{u}, x_0)\}_{i=0}^{T'}$ but assume access to future controls. Then the prediction objective for a class of systems for $N$ examples for timesteps $\{t_i\}_{T'+1}^T$ is

$$\hat{x} = \arg\min_{\hat{x}} \mathbb{E}_{\phi \sim P(\Phi), x_0 \sim P(X_0)} \left[ \sum_{i=T'+1}^T ||x(\phi, t_i, \mathbf{u}, x_0) - \hat{x}_{t_i}||_2^2 \right]. \tag{1}$$

This objective differs from the traditional one in that implicitly, identifying $\phi$ for each trajectory needs to be done from the problem data in order to be able to predict the data generated by $f_\phi$.

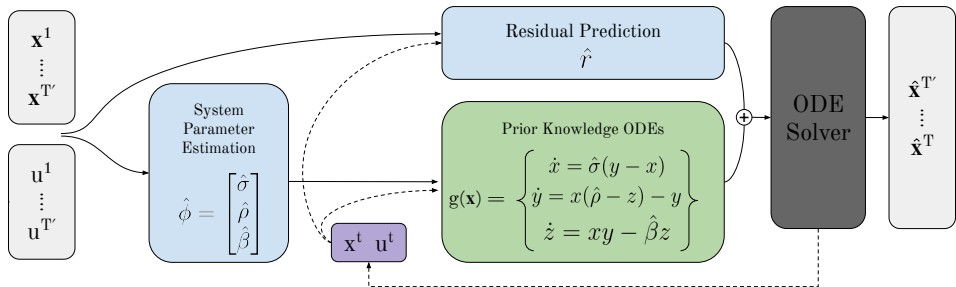

Figure 1: **An example Neural Dynamical System.** Here, blue boxes are fully connected neural networks, gray boxes are problem data and output, the green box is the prior knowledge dynamical system, the purple box is data output by ODE solver to query derivatives, and the black box is an ODE solver. The ODEs and system parameters are problem dependent, but here we consider the Lorenz system (defined in Example 1) as an example. Our notation for $x$ is unfortunately overloaded by our method and the Lorenz system—the $x$ from our method is bolded in the figure.

Similarly, the control problem is

$$\mathbf{u} = \min_{\mathbf{u}} \mathbb{E}_{\phi \sim P(\Phi), x_0 \sim P(X_0)} \left[ \int_0^t c(u(t), x(t)) dt \right], \text{ s.t. } x(t) = x_0 + \int_0^t f_\phi(x, u, t) \, dt \quad (2)$$

for some cost functional $c$. We will primarily explore the prediction problem in this setting, but as secondary considerations, we explore robustness to noise, the ability to handle irregularly spaced input data, and the ability to recover the parameters $\phi$ which generated the original trajectories. We will also consider the control problem in a simple setting.

## 4 METHODS

We build up the description of our proposed method by first describing the two methods that inspire it: gray box system identification through optimization (Ljung et al., 2009), and using a Neural ODE (Chen et al., 2018) to predict future states in a dynamical system.

To apply grey box optimization (Ljung et al., 2009) to a dynamical system $\dot{x} = f_\phi(x, u, t)$ for problem data $\{x_{t_i}\}_{t=0}^{T'}$, we would use nonlinear least squares (Coleman & Li, 1996) to find

$$\hat{\phi} = \arg\min_{\hat{\phi}} \sum_i \left\| \int_0^{t_i} f_\phi(x, u(t), t) \, dt - \int_0^{t_i} f_{\hat{\phi}}(x, u(t), t) dt \right\|. \quad (3)$$

This makes good use of the prior knowledge component of our system but is prone to compounding errors through integration and does not leverage data that may have come from alternate system parameters.

A data driven approach would be to minimize the same objective with a fully connected neural ODE (Chen et al., 2018) $h_\theta$ in place of $f$. However, we find that this procedure requires large amounts of training data and doesn't leverage any prior knowledge we might have, though it is flexible to classes of dynamical systems.

We define a Neural Dynamical System by taking the advantages of both these methods in the setting where we know the full and correct ODEs and then show how to generalize it to situations where only some ODEs are known or they are approximate. Specifically, a *Neural Dynamical System* (NDS) is a class of dynamical systems where a neural network predicts some part of $f_\phi(x, u, t)$, usually parameters $\phi$ or a term which is added to $f$.

**NDS with Full System Dynamics** Consider a class of dynamical systems as defined in Section 3 where $x \in \mathbb{R}^n$, $u \in \mathbb{R}^m$, $\phi \in \mathbb{R}^{d_p}$, $d_h, d_c \in \mathbb{N}$ and let $\theta, \vartheta, \tau$ be trainable neural network weights. Let $h_\theta(x_{t_{1:T'}}, u_{t_{1:T}})$ be a neural net mapping state history and control sequence to the $d_p$ parameters of the system $\hat{\phi}$ and an embedding $b_h \in \mathbb{R}^{d_h}$. Also let $c_\vartheta(x_t, u_t)$ be a similar network taking a single state and control that outputs an embedding $b_c \in \mathbb{R}^{d_c}$. Finally, let $d_\tau(b_h, b_c)$ be a network which takes the two output embeddings from the previous network and outputs residual terms $\hat{r}$. Intuitively, we would like to use the observed history to estimate our system parameters, and some combination

of the observed history and current observation to estimate residuals, which influences the design of our model, the neural dynamical system (a visualization of which is shown in Figure 1), written

$$\dot{x} = \underbrace{g_{\hat{\phi}}(x_t, u_t, t)}_{\text{Prior knowledge}} + \hat{r} \quad \hat{\phi}, b_h = \underbrace{h_\theta(x_{t_{1:T'}}, u_{1:T})}_{\text{History encoder}} \quad b_c = \underbrace{c_\vartheta(x_t, u_t)}_{\text{Context encoder}} \quad \hat{r} = \underbrace{d_\tau(b_h, b_c)}_{\text{Residual prediction}} \tag{4}$$

where $g$ are domain-specific ODEs which are the input 'domain knowledge' about the system being modeled. Note that if the prior knowledge $g$ is identically zero, this method reduces to the Neural ODE predictions we discussed at the beginning of this section. We also study an ablated model, NDS0, which lacks the residual component $\hat{r}$ and context encoder network $d_\tau$. We note here that the context encoder is intended to potentially correct for model misspecification and noise but in the noiseless case with a model which is perfect, it may not be necessary. We explore this throughout Section 5.

*Example 1: Lorenz system.* To illustrate the full construction, we operate on the example of the the Lorenz system: a chaotic dynamical system originally defined to model atmospheric processes (Lorenz, 1963). The system has 3-dimensional state (which we'll denote by $x, y, z$), 3 parameters, $\rho, \sigma$, and $\beta$, and no control input. The system is given by

$$\dot{x} = \sigma(y - x) \quad \dot{y} = x(\rho - z) - y \quad \dot{z} = xy - \beta z. \tag{5}$$

For a given instantiation of the Lorenz system, we have values of $\phi = [\beta, \sigma, \rho]$ that are constant across the trajectory. So, we can instantiate $h_\theta$ which outputs $\hat{\phi} = [\hat{\beta}, \hat{\sigma}, \hat{\rho}]$. We use the DOPRI5 method (Dormand & Prince, 1980) to integrate the full neural dynamical system in Equation 4, with $g$ given by the system in Equation 5 using the adjoint method of Chen et al. (2018). We use the state $x_{T'}$ as the initial condition for this integration. This gives a sequence $\{\hat{x}_t\}_{t=T'}^T$, which we evaluate and supervise with a loss of the form

$$\mathcal{L}_{\theta, \vartheta, \tau}(\{\hat{x}_{t_i}\}_{i=T'+1}^T, \{x_{t_i}\}_{t=T'+1}^T) = \sum_{t=T'+1}^T ||x_{t_i} - \hat{x}_{t_i}||_2^2. \tag{6}$$

Because of the way we generate our input data, this is equivalent to Equation 1. We assume in our setting with full dynamics that the true dynamics lie in the function class established in Equation 4. By the method in Chen et al. (2018) we can backpropagate gradients through this loss into the parameters of our NDS. Then algorithms in the SGD family will converge to a local minimum of our loss function.

**NDS with Partial System Dynamics** Suppose we only had prior knowledge about some of the components of our system and none about others. We can easily accomodate this incomplete information by simply 'zeroing out' the function. This looks like

$$g_\phi(x, u, t) = \left[ \begin{cases} g_\phi^{(i)}(x, u, t) & \text{if } g_\phi^{(i)} \text{ is known,} \\ 0 & \text{else.} \end{cases} \right] \tag{7}$$

substituted into equation 4. In this setup, the residual term essentially makes the unknown dimensions unstructured Neural ODEs, which still can model time series well (Portwood et al., 2019).

**NDS with Approximate System Dynamics** For Neural Dynamical Systems to be useful, they must handle situations where the known model is approximate. This is transparently handled by our formulation of Neural Dynamical Systems: the parameters of the approximate model $\hat{\phi}$ are predicted by $h_\theta(x_{1:T'}, u_{1:T'})$ and the residuals $\hat{r}$ are predicted by $d_\tau(b_h, b_c)$. This is the same as in the case where we have the correct dynamics, but we remove the assumption of a perfect model.

*Example 2: Nuclear Fusion System.* In this paper, we apply this technique to plasma dynamics in a tokamak. In a tokamak, two quantities of interest are the stored energy of the plasma, which we denote $E$ and its rotational frequency, $\omega$. The neutral beams and microwave heating allow us to add power ($P$) and torque ($T$) to the plasma. The plasma also dissipates energy and rotational momentum via transport across the boundary of the plasma, radiative cooling, and other mechanisms. While the detailed evolution of these quantities is described by turbulent transport equations, for the purposes of control and design studies, physicists often use reduced, volume-averaged models. The

simple linear model (up to variable parameters) used for control development in Boyer et al. (2019a) is used in this work.

$$\dot{E} = P - \frac{E}{\tau_e} \quad \dot{\omega} = \frac{T}{n_i m_i R_0} - \frac{\omega}{\tau_m} \tag{8}$$

Here, $n_i$ is ion density, $m_i$ is ion mass, and $R_0$ is the tokamak major radius (values are in A.4). We use the constant known values for these. $\tau_e$ and $\tau_m$ are the confinement times of the plasma energy and angular momentum, which we treat as variable parameters (because they are!). These are predicted by the neural network in our model. We again use the model in Equation 4 to give us a neural dynamical system which can learn the real dynamics starting from this approximation in Section 5.2.

## 5 EXPERIMENTS

In the following experiments, we aim to show that our methods improve predictions of physical systems by including prior dynamical knowledge. These improvements hold even as we vary the configurations between structured and fluid settings. We show that our models learn from less data and are more accurate, that they handle irregularly spaced data well, and that they learn the appropriate parameters of the prior knowledge systems even when they only ever see trajectories.

We use L2 error as our evaluation measure for predictive accuracy as given by Equation 6. We also evaluate our model's ability to predict the system parameters by computing the L2 error, i.e. $\sum_{i=1}^{n} ||\hat{\phi}_i - \phi_i||_2^2$. In the settings where we are adding either noise or, as will be defined, 'jitter', we use percent change in L2 error of the trajectories relative to a noise- or jitter-free baseline for the same experiment. We believe this is the appropriate metric as it abstracts away the original differences in accuracy between the methods and focuses on the effects of the noise or jitter.

For synthetic examples, we consider the Lorenz system in equation 5 and the Ballistic system (A.1.1). We learn over trajectories $\{(x_{t_i}, u_{t_i}, t_i)\}_{i=1}^{T}$ where the $x_{t_i}$ are generated by numerically integrating $\dot{x}_\phi(x, u, t)$ using scipy's odeint function (Virtanen et al., 2019), with $x_0$ and $\phi$ uniformly sampled from $\mathcal{X}$ and $\Phi$, and $u_{t_i}$ given. Note that $\phi$ remains fixed throughout a single trajectory. Details on the ranges of initial conditions and parameters sampled are in the appendix. We evaluate the synthetic experiments on a test set of 20,000 trajectories that is fixed for a particular random seed generated in the same way as the training data. We use a timestep of $0.5$ seconds for the synthetic trajectories. On the Ballistic system this allows us to see trajectories that do not reach their peak and those that start to fall. Since the Lyapunov exponent of the Lorenz system is less than 3, in 16 predicted timesteps we get both predictable and unpredictable data (Frøyland & Alfsen, 1984)). We believe it is important to look at the progress of the system across this threshold to understand whether the NDS model is robust to chaotic dynamics — since the Lorenz system used for structure is itself chaotic, we want to make sure that the system does not blow up over longer timescales.

We compare our models with other choices along the spectrum of structured to flexible models from both machine learning and system identification. The models we looked at for experiments are the Full NDS, a Partial NDS, a NDS0, a fully connected neural network (FC), a fully connected neural ODE (FC NODE), an LSTM, MATLAB's Gray Box Optimization (GBO), and a sparse regression algorithm (SR) due to Brunton et al. (2015). Details on each algorithm are given in Appendix A.2.

We can view the Partial NDS and NODE as ablations of the Full NDS model which remove some and all of the prior knowledge, respectively. Each model takes 32 timesteps of state and control information as input and are trained on predictions for the following 16 timesteps. The ODE-based models are integrated from the initial conditions of the last given state. All neural networks are all trained with a learning rate of $3 \times 10^{-3}$, which was seen to work well across models. We generated a training set of 100,000 trajectories, test set of 20,000 trajectories, and validation set of 10,000 trajectories. Training was halted if validation error did not improve for 3 consecutive epochs.

### 5.1 SYNTHETIC EXPERIMENTS

We first present results on a pair of synthetic physical systems where the data is generated in a noiseless and regularly spaced setting. We then add noise (in A.9) and irregular spacing to our data to highlight the performance of these methods as conditions become more challenging.

**Sample Complexity and Overall Accuracy** In order to test sample complexity in learning or fitting, we generated data for a full training dataset of 100,000 trajectories. We then fit our models on different fractions of the training dataset: $1, 0.25, 0.05, 0.01$. We repeated this process with 5

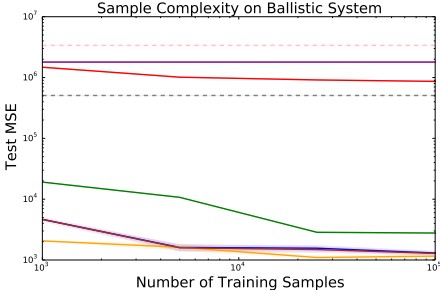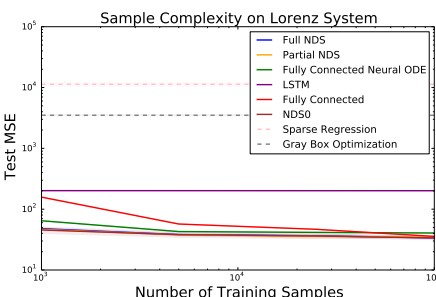

Figure 2: **L2 loss between predicted and real trajectory as we train on more samples.** The NDS models learn with much fewer samples and converge to much lower errors on both systems. Shaded regions are standard errors.

different random seeds and plotted the $L2$ error of the model over the dataset fraction seen by the model in Figure 2. The error regions are the standard error of the errors over the various seeds.

As seen in Figure 4, the learning of Neural Dynamical Systems looks very different to that of the comparison models. We also see that with small amounts of data, the NDS models greatly outperform the Neural ODE, but with the full dataset, their performances get closer. This makes sense as the Neural ODE is likely able to infer the structure of the system with large amounts of data. Also, the Fully Connected Neural ODE outperforms the other baselines, which we posit is due to the fact that it implicitly represents that this system as a continuous time dynamical process and should change in a continuous fashion. From a sample-complexity perspective it makese sense that the better initialization of NDS should matter most when data is limited. A table of the full results of all experiments can be seen in A.11.

We notice that the NDS0 slightly outperforms the NDS with higher variance on these systems. Since it has a perfect model of the system, the residual components aren't necessary for the model to perform well, however, there is no way the network can 'correct' for a bad estimate.

Curiously, we see on the ballistic system that the partial NDS outperforms the full NDS in the small data setting, but they converge to similar performance with slightly more data. A potential explanation for this is that errors propagate through the dynamical model when the parameters are wrong, while the partial systems naturally dampen errors since, for example, $\dot{z}$ only depends on the other components through a neural network. Concretely this might look like a full NDS predicting the wrong Rayleigh number $\sigma$ which might give errors to $y$ which would then propagate to $x$ and $y$. Conversely, this wouldn't happen as easily in a partial NDS because there are neural networks intermediating the components of the system.

**Noise, Irregular Sampling, and Parameter Identification**    We include further experiments on irregular sampling in Figure 6, where we see that the NDS models are the only ones which do not suffer from varied spacing between timesteps. The Neural ODE model does modestly worse, while the feedforward and recurrent machine learning models struggle. We hypothesize that the initialization of the NDS models helps it generalize to this setting as the prior knowledge models are necessarily invariant to the timestep.

We also include experiments with the performance of our methods under uniform noise in Figure 7. NDS models may be less robust to noise than the machine learning and system identification competitors since they experience a greater relative change in performance (however they still exhibit small absolute changes in performance). They still outperform the baselines on an absolute basis. We hypothesize that here the relatively inflexible prior knowledge causes errors to compound in a way that is only somewhat mitigated by the residual terms.

Additionally, we tested the subset of models which identify model parameters to see how close they are to the true values in Figure A.6. Here, we see that the NDS models do well but slightly worse than NDS0. We also see that the NDS models benefit from the generalization of the neural network parameter estimation over the GBO which does a new optimization per trajectory.

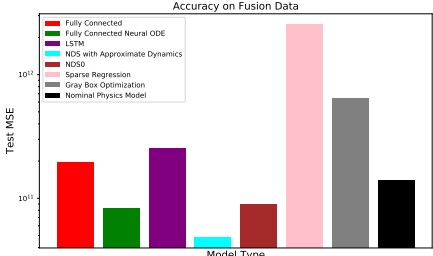
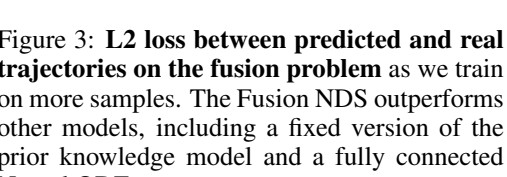
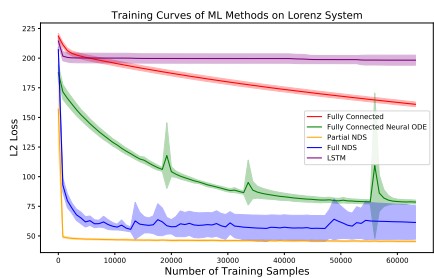

Figure 3: **L2 loss between predicted and real trajectories on the fusion problem** as we train on more samples. The Fusion NDS outperforms other models, including a fixed version of the prior knowledge model and a fully connected Neural ODE.

Figure 4: A typical learning curve for NDS. We believe the quick drop in error seen in these models can be thought of as a consequence of a much better initialization in function space.

## 5.2 FUSION EXPERIMENTS

We explored the concept of approximate system dynamics in a simplified fusion system. We predict the state of the tokamak as summarized by its stored energy and rotational frequency given the time series of control input in the form of injected power and torque. As mentioned in Section 4, we have a simplified physical model given by Equation 8 that approximately gives the dynamics of these quantities and how they relate to one another through time. Though there is a lot of remaining work to apply this model in a real experiment, approaches merging theoretical models with data to make useful predictions can be embedded into useful controller designs and improve the state of fusion.

Our full dataset consisted of 17,686 trajectories, which we randomly partitioned into 1000 as a test set and 16,686 as a training set.[1] The data are measured from the D-IIID tokamak via magnetic reconstruction (Ferron et al., 1998) and charge-exchange recombination spectroscopy (Haskey et al., 2018). Similar to our synthetic experiments, we cut each trajectory into overlapping 48 timestep sections and train on 32 timesteps to predict 16 timesteps. We compare with the same models as in the previous section, but using our Fusion Neural Dynamical System as described in Equation 4 with $g$ given by Equation 8. As we discussed above, the dynamics in this equation are approximate. To illustrate this, we have included the accuracy of the naive dynamics with no learning on our data with fixed confinement times $\tau_e = \tau_m = 0.1$s as the Nominal Fusion Model in Figure 3. We use a larger fully connected network with 6 layers with 512 hidden nodes to attempt to capture the added complexity of the problem.

**Sample Complexity and Overall Accuracy** When comparing our NDS models, the machine learning baselines, the system ID baselines, and a nominal model from Boyer et al. (2019b), we see that the Fusion NDS model performs best by a large margin. Although the fully connected neural ODE performs competitively, it fails to reach the same performance. We speculate that the dynamical model helps with generalization whereas the fully connected network may overfit the training data and fail to reach good performance on the test set. Here the NDS0 is unable to perform well compared to the NDS, as the approximate dynamics mean that the model error is somewhat catastrophic for predictions. We see however that the NDS0 outperforms the Nominal Physics Model as it is able to estimate the parameters for each example rather than fixing values of the parameters for the whole dataset.

We see these results as highly encouraging and will continue exploring uses of NDS in fusion applications.

---

[1]Data is loaded and partially processed within the OMFIT framework (Meneghini et al., 2015). We used the "SIGNAL_PROCESSING" module which has recently been developed for this task and is publicly available on the "profile_prediction_data_processing" branch of the OMFIT source code. Details of the preprocessing are in the Appendix.

| | MSE of Model | | Performance of MPC | |
|---|---|---|---|---|
| # of Training Examples | 5K | 1K | 5K | 1K |
| FC | $0.031 \pm 0.009$ | $0.058 \pm 0.018$ | $52 \pm 3$ | $41 \pm 4$ |
| FC NODE | $0.028 \pm 0.011$ | $0.049 \pm 0.013$ | $55 \pm 4$ | $46 \pm 3$ |
| LSTM | $0.081 \pm 0.023$ | $0.092 \pm 0.025$ | $23 \pm 6$ | $25 \pm 8$ |
| Full NDS | $\mathbf{0.020 \pm 0.006}$ | $\mathbf{0.029 \pm 0.007}$ | $\mathbf{72 \pm 4}$ | $\mathbf{60 \pm 3}$ |
| Partial NDS | $\mathbf{0.022 \pm 0.009}$ | $\mathbf{0.033 \pm 0.011}$ | $\mathbf{69 \pm 8}$ | $55 \pm 6$ |
| NDS0 | $\mathbf{0.023 \pm 0.013}$ | $\mathbf{0.028 \pm 0.014}$ | $\mathbf{71 \pm 11}$ | $\mathbf{57 \pm 8}$ |
| SR | $0.037 \pm 0.023$ | $0.041 \pm 0.015$ | $65 \pm 4$ | $56 \pm 4$ |
| GBO | $0.046 \pm 0.019$ | n/a | $49 \pm 5$ | n/a |

Table 1: Modeling and Control on the EvilCartpole system.

## 5.3 CONTROL EXPERIMENT

We also explored the use of these models for control purposes using model-predictive control (Camacho & Alba, 2013). For this purpose, we modified the Cartpole problem from Brockman et al. (2016) so that there are a variety of parameter values for the weight of the cart and pole as well as pole length as specified in Section A.1.2. Typically, a 'solved' cartpole environment would imply a consistent performance of 200 from a control algorithm. However, there are three factors that make this problem more difficult. First, in order to allow each algorithm to identify the system in order to make appropriate control decisions, we begin each rollout with 8 random actions. The control never fails at this point but would certainly fail soon after if continued. Second, the randomly sampled parameters per rollout make the actual control problem more difficult as the environment responds less consistently to control. For example, MPC using the typical Cartpole environment as a model results in rewards of approximately 37. Third, all training data for these algorithms uses random actions with no exploration, which has been seen to degrade the performance of most model-based RL or control algorithms (Mozer & Bachrach, 1990).

We then trained dynamics models on this 'EvilCartpole' enviroment for each of our comparison algorithms on datasets of trajectories on the environment with random actions. At that point, we rolled out trajectories on our EvilCartpole environment using MPC with control sequences and random shooting with 1,000 samples and a horizon of 10 timesteps. The uncertainties are standard errors over 5 separately trained models.

As shown in Table 1, the NDS algorithms outperform all baselines on the cartpole task for both the modeling and control objectives. We see that all algorithms degrade in performance as the amount of data is limited. We see in Table 6 that with larger amounts of data the Fully Connected and Neural ODE models perform as well as the NDS models. We hypothesize that this is due to the fact that the cartpole dynamics are ultimately not that complicated and with sufficient data unstructured machine learning algorithms can learn the appropriate dynamics to reach a modestly performing controller as well as NDS.

## 6 CONCLUSION

In conclusion, we give a framework that merges theoretical dynamical system models with deep learning by backpropagating through a numerical ODE solver. This framework succeeds even when there is a partial or approximate model of the system. We show there is an empirical reduction in sample complexity and increase in accuracy on two synthetic systems and on a real nuclear fusion dataset. In the future, we wish to expand upon our work to make more sophisticated models in the nuclear fusion setting as we move toward practical use. We also hope to explore applications of this framework in other area which have ODE-based models of systems.

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

## A APPENDIX

### A.1 ADDITIONAL DYNAMICAL SYSTEMS USED

#### A.1.1 BALLISTIC SYSTEM

We also predict trajectories for ballistics: an object is shot out of a cannon in the presence of air resistance. It has a mass and drag coefficient and follows a nearly parabolic trajectory. This system has a two-dimensional state space (altitude $y$ and horizontal range $x$) and 2 parameters (mass and drag coefficient), which we reduce down to one: terminal velocity $v_t$. It is a second order system of differential equations which we reduce to first order using the standard refactoring.

The system is given by

$$
\begin{aligned}
\dot{x} = \dot{x} \quad & \ddot{x} = -\frac{g\dot{x}}{v_t} \\
\dot{y} = \dot{y} \quad & \ddot{y} = -g\left(1 + \frac{\dot{y}}{v_t}\right)
\end{aligned}
\tag{9}
$$

(here, $g$ is the constant of gravitational acceleration).

#### A.1.2 CARTPOLE SYSTEM

In section 5.3 we discuss experiments on a modified Cartpole system with randomly sampled parameters. Here we give a full delineation of that system as defined in Brockman et al. (2016) with our modifications.

This system has three parameters, one control, and four state variables. The parameters are $l$, the length of the pole, $m_c$, the mass of the cart, and $m_p$, the mass of the pole. The control is $F$, the horizontal force applied to the cart. In the current setup, $u$ can be one of $\pm 10$. The state variables are the lateral position $x$, the angle of the pole from vertical $\theta$, and their first derivatives, $\dot{x}$ and $\dot{\theta}$.

The system is given by the following equations

$$
\begin{aligned}
\dot{\theta} = \dot{\theta} \quad & \ddot{\theta} = \frac{g\sin\theta + \cos\theta\left(\frac{-F - m_p l\dot{\theta}^2 \sin\theta}{m_c + m_p}\right)}{l\left(\frac{4}{3} - \frac{m_p\cos^2\theta}{m_c + m_p}\right)} \\
\dot{x} = \dot{x} \quad & \ddot{x} = \frac{F + m_p l(\dot{\theta}^2 \sin\theta - \ddot{\theta}\cos\theta)}{m_c + m_p}
\end{aligned}
\tag{10}
$$

We give a reward of $+1$ for every timestep that $|\theta| \leq \frac{\pi}{15}$ and $|x| \leq 2.4$. Initial conditions are uniformly sampled from $[-0.05, 0.05]^4$ at test time and at training $x \sim U([-2, 2])$, $\dot{x} \sim U([-1, 1])$, $\theta \sim U([-0.2, 0.2])$ (which is slightly wider than the $\frac{\pi}{15}$ threshold used at test), and $\dot{\theta} \sim U([-0.1, 0.1])$. The parameters are also uniformly sampled at train and test with $l \sim U([0.6, 1.2])$, $m_c \sim U([0.5, 2])$ and $m_p \sim U([0.03, 0.2])$.

For the partial NDS for Cartpole, we remove the equations corresponding to $\dot{x}$ and $\ddot{x}$.

### A.2 COMPARISON METHODS

In our paper, we compared the following methods in our experiments:

- **Full NDS**: A Neural Dynamical System with the full system dynamics for the problem being analyzed. The full construction of this model is given by Equation 4. For the functions $h_\theta, c_\vartheta, d_\tau$, we use fully connected networks with 2 layers, Softplus activations, 64 hidden nodes in each layer, and batch normalization.

- **Partial NDS**: A Neural Dynamical System with partial system dynamics for the problem being analyzed. These follow Equation 7 as applied to Equation 4. For the Ballistic system, we only provide equations for $\dot{x}$ and $\ddot{x}$, excluding the information about vertical motion from our network. For the Lorenz system, we only provide equations for $\dot{x}$ and $\dot{y}$, excluding information about motion in the $z$ direction. For the Cartpole system, we only provide information about $\dot{\theta}$ and

$\ddot{\theta}$. These equations were chosen somewhat arbitrarily to illustrate the partial NDS effectiveness. We use similar neural networks here as for the Full NDS.

- **NDS0**: A Full NDS with residual terms removed. This serves as an ablation which shows the use of the residual terms.

- **Fully Connected** (FC): A Fully-Connected Neural Network with 4 hidden layers containing 128 nodes with ReLU activations and batch normalization.

- **Fully Connected Neural ODE** (FC NODE): A larger version of the Neural ODE as given in Chen et al. (2018), we use 3 hidden layers with 128 nodes, batch norm, and Softplus activations for $\dot{x}$. This can be interpreted as a version of our NDS with no prior knowledge, i.e. $g(x) = 0$.

- **LSTM**: A stacked LSTM with 8 layers as in Graves (2013). The data is fed in sequentially and we regress the outputs of the LSTM against the true values of the trajectory.

- **Gray Box Optimization** (GBO): We use MATLAB's gray-box system identification toolbox (Ljung et al., 2009) along with the prior knowledge ODEs to fit the parameters $\hat{\phi}$ as an alternative to using neural networks. This algorithm uses trust-region reflective nonlinear least squares with finite differencing (Coleman & Li, 1996) to find the parameter values which minimize the error of the model rollouts over the observed data.

- **Sparse Identification of Nonlinear Systems** (SR): We use the method from Brunton et al. (2015) to identify the dynamical systems of interest. This method uses sparse symbolic regression to learn a linear mapping from basis functions of the state $x_t$ and control $u_t$ to the derivatives $\dot{x}_t$ computed by finite differences. Our synthetic systems are in the span of the polynomial basis that we used.

We note that ReLU activations were chosen for all feedforward and recurrent architectures, while in the Neural-ODE-based architectures, we follow the recommendations of Chen et al. (2018) and use the Softplus. The sizes and depths of the baselines were chosen after moderate hyperparameter search.

## A.3 HYPERPARAMETERS

We trained using Adam with a learning rate of $3 \times 10^{-3}$ and an ODE relative and absolute tolerance when applicable of $10^{-3}$. This wide tolerance was basically a function of training time as with tighter tolerances experiments took a long time to run and we were more concerned with sample complexity than the tightness of the integration. Hyperparameters were predominantly chosen by trial and error.

Over the course of figuring out how this works and then evaluating the models we were evaluating, we ran some form of this code 347 times. The typical setup was either a 1080Ti GPU and 6 CPU cores or 7 CPU cores for roughly a day per experiment. We found that most of these trainings were marginally faster on the GPU  1.5x speedup, but weren't religious about the GPU as we had many CPU cores and many experiments to run in parallel.

**Lorenz Initial Conditions and Parameters**    For our Lorenz system, we sampled $\rho \sim U([15, 35])$, $\sigma \sim U([9, 12])$, $\beta \sim U([1, 3])$, $x_0 \sim U([0, 5])$, $y_0 \sim U([0, 5])$, $z_0 \sim U([0, 5])$.

**Ballistic Initial Conditions and Parameters**    For our Ballistic System, we sampled masses $m \sim U([1, 100])$, drag coefficients $c_d \sim U([0.4, 3])$, $x_0 \sim U([-100, 100])$, $y_0 \sim U([0, 200])$. We then use $v_t = \frac{mg}{c_d}$ to recover the terminal velocity used in our model.

## A.4 FUSION MODEL PARAMETERS

$n_i$ is ion density (which we approximate as a constant value of $5 \times 10^{19}$ deuterium ions per $m^3$), $m_i$ is ion mass (which we know since our dataset contains deuterium shots and the mass of a deuterium ion is $3.3436 \times 10^{-27}$ kg), and $R_0$ is the tokamak major radius of 1.67 m.

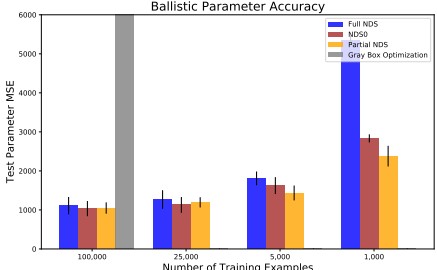 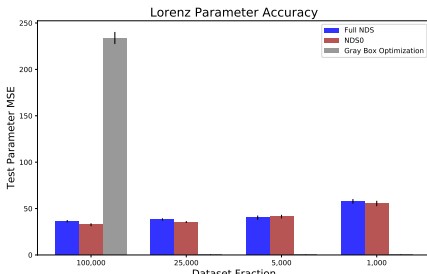

Figure 5: $L2$ **distance between** $\phi$ **and** $\hat{\phi}$**.** As the NDS are trained under the usual $L2$ supervision, the parameters $\hat{\phi}$ of the system approach the correct values.

## A.5 FUSION DATA PREPROCESSING

Data is loaded and partially processed within the OMFIT framework Meneghini et al. (2015). A new module, "SIGNAL_PROCESSING", has been developed for this task and is publicly available on the "profile_prediction_data_processing" branch of the OMFIT source code. The rest of the processing is done on Princeton's Traverse computing cluster, and is available in the GitHub sourcecode for this project (https://github.com/jabbate7/plasma-profile-predictor).

DIII-D shots from 2010 through the 2019 campaign are collected from the MDS+ database. Shots with a pulse length less than 2s, a normalized beta less than 1, or a non-standard topology are excluded from the start. A variety of non-standard data is also excluded, including the following situations:

1. during and after a dudtrip trigger (as determined by the pointname "dustripped")
2. during and after ECH (pointname "pech" greater than .01) activation, since ECH is not currently included as an actuator
3. whenever density feedback is off (as determined by the pointname "dsifbonoff")
4. during and after non-normal operation of internal coil, with an algorithm described by Carlos Paz-Soldan

All signals are then put on the same 50ms time base by averaging all signal values available between the current time step and 50ms prior. If no data is available in the window, the most recent value is floated. Only time steps during the "flattop" of the plasma current are included. The flattop period is determined by DIII-D's "t_ip_flat" and "ip_flat_duration" PTdata pointnames.

All profile data is from Zipfit Meneghini et al. (2015). The profiles are linearly interpolated onto 33 equally spaced points in normalized toroidal flux coordinates, denoted by $\rho$, where $\rho = 0$ is the magnetic axis and $\rho = 1$ is the last closed flux surface.

## A.6 PARAMETER LEARNING WITHOUT EXPLICIT SUPERVISION

For experiments in Figure 2, we stored the parameter estimates $\hat{phi}$ for the NDS and gray box models and compared them to the true values to see how they perform in identification rather than prediction. None of these models were ever supervised with the true parameters. We see in Figure A.6 that the NDS is better able to estimate the parameter values than the gray-box method for both systems tested. We believe this is because our method is able to leverage many trajectories to infer the parameters whereas the gray-box method only uses the single trajectory.

## A.7 COMPUTATION OF NOISE

In our previous experiments, we did not add noise to the trajectories generated by the synthetic systems. We generate noise added to the trajectories by first sampling a set of 100 trajectories and computing for the $i$th component of the state space the RMS value $c_i$ across our 100 trajectory

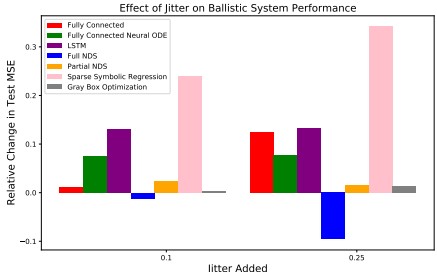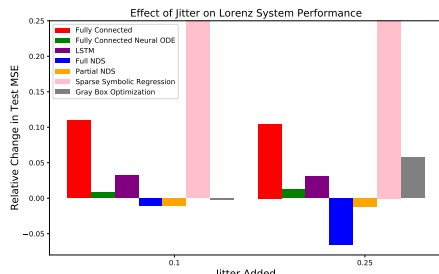

Figure 6: **Change in performance when data are irregularly sampled in time.** The Full NDS performs better than any comparison model under added jitter.

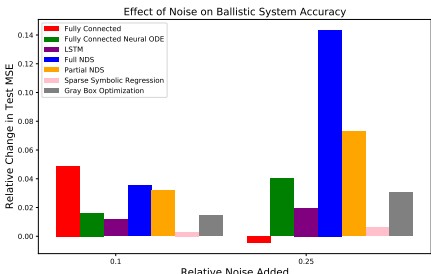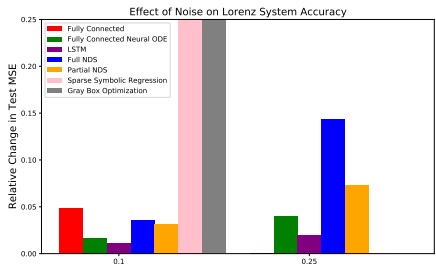

Figure 7: **Change in performance under added relative noise $r$.** The NDS seem to have more variance in performance under high amounts of noise as the comparison models but it still greatly outperforms them on an absolute basis.

samples. Then we sample noise $\mathcal{N}(0, rc_i)$ from a normal distribution where the variance is $c_i$ scaled by our 'relative noise' term $r$. We vary $r$ to control the amount of noise added to a system in a way that generalizes across problems and across components of a problem's state space.

## A.8 ROBUSTNESS TO IRREGULAR TIMESTEPS

We also explored how these models respond to data sampled in intervals that are not regularly spaced in a fashion discussed in A.10. In Figure 6, we see that the ODE-based models are able to handle the irregularly spaced data much better than the others. Like with noise, we are focusing on relative performance but even so, the full NDS even does substantially better under large jitter settings. As we have discussed, this makes sense because these models natively handle time. We conjecture that the Full NDS neural networks may learn something slightly more general by this 'domain randomization' given that they are correctly specified models that receive all the information of the differing timesteps. The symbolic regression method fails under jitter, presumably because it relies heavily on finite differencing.

## A.9 ROBUSTNESS TO ADDED NOISE

We evaluate our experiments on additive noise scaled relative to the system as discussed in A.7. As the NDS does substantially better than the other models on the synthetic data, we look at the effect of noise relative to the original performance to focus on the properties of the new model. When a large amount of noise is added to the system, NDS's performance degrades faster than that of the other models, though it still outperforms them on an absolute basis. We think full and partial NDS might be unstable here due to errors propagating through the prior knowledge model. The other models all otherwise perform fairly similarly under added noise.

| System | Lorenz | | | | Ballistic | | | |
|---|---|---|---|---|---|---|---|---|
| Samples | 100,000 | 25,000 | 5,000 | 1,000 | 100,000 | 25,000 | 5,000 | 1,000 |
| FC | $35.7 \pm 0.1$ | $46.7 \pm 0.2$ | $57.0 \pm 0.2$ | $157.9 \pm 0.5$ | $864K \pm 3K$ | $912K \pm 4K$ | $1013K \pm 3K$ | $1477K \pm 5K$ |
| FC NODE | $40.5 \pm 0.6$ | $41.5 \pm 0.6$ | $42.9 \pm 0.5$ | $64.5 \pm 0.5$ | $3K \pm 17$ | $3K \pm 31$ | $10K \pm 58$ | $19K \pm 81$ |
| Full NDS | $\mathbf{33.8 \pm 2}$ | $36.6 \pm 2$ | $38.3 \pm 2$ | $47.8 \pm 2$ | $1.3K \pm 90$ | $1.6K \pm 160$ | $1.6K \pm 160$ | $4.6K \pm 35$ |
| Partial NDS | $34.8 \pm 1$ | $\mathbf{35.8 \pm 2.2}$ | $\mathbf{37.8 \pm 1.1}$ | $\mathbf{46.7 \pm 1.1}$ | $\mathbf{1.2K \pm 100}$ | $\mathbf{1.1K \pm 150}$ | $\mathbf{1.6K \pm 190}$ | $\mathbf{2.1K \pm 210}$ |
| NDS0 | $\mathbf{33.7 \pm 2.2}$ | $36.2 \pm 7.3$ | $38.1 \pm 8.3$ | $45.5 \pm 12.5$ | $\mathbf{1.3K \pm 143}$ | $1.5K \pm 347$ | $1.6K \pm 434$ | $5.7K \pm 743$ |
| LSTM | $198 \pm 4$ | $200 \pm 6$ | $201 \pm 5$ | $201 \pm 4$ | $180K \pm 2K$ | $181K \pm 3K$ | $181K \pm 3K$ | $181K \pm 3K$ |
| SR | $3.7K \pm 6$ | n/a | n/a | n/a | $2.1M \pm 120K$ | $2.8M \pm 170K$ | $3.1M \pm 140K$ | $3.1M \pm 180K$ |
| GBO | $3.2K \pm 200$ | n/a | n/a | n/a | $8.2K \pm 600$ | n/a | n/a | n/a |

Table 2: Sample Complexity Results as discussed in Section 5.1 and Figure 2.

| System | Lorenz | | | | Ballistic | | | |
|---|---|---|---|---|---|---|---|---|
| Samples | 100,000 | 25,000 | 5,000 | 1,000 | 100,000 | 25,000 | 5,000 | 1,000 |
| Full NDS | $36.4 \pm 2.8$ | $38.3 \pm 2.8$ | $40.3 \pm 4.9$ | $57.8 \pm 5.5$ | $1110 \pm 497$ | $1267.81 \pm 531$ | $\mathbf{1308.9 \pm 394}$ | $\mathbf{2349 \pm 278}$ |
| Partial NDS | n/a | n/a | n/a | n/a | $1249 \pm 324$ | $\mathbf{1194 \pm 293}$ | $1434.8 \pm 426$ | $2378 \pm 593$ |
| NDS0 | $\mathbf{32.5 \pm 3.8}$ | $\mathbf{35.4 \pm 4.3}$ | $\mathbf{41.3 \pm 4.3}$ | $\mathbf{55.4 \pm 6.4}$ | $\mathbf{1034 \pm 434}$ | $1128 \pm 453$ | $1624 \pm 484$ | $2833 \pm 233$ |
| GBO | $233.8 \pm 15$ | n/a | n/a | n/a | $160990 \pm 1261$ | n/a | n/a | n/a |

Table 3: Parameter Error Results as discussed in Section 5.1 and Figure A.6.

## A.10 Computation of Jitter

Though, as we will discuss, our fusion data in section 5.2 has been postprocessed to be regularly spaced, in practice, data on tokamaks and in many other systems come in irregularly and are sometimes missing. As both the fully connected neural ODE and NDS models are integrated in continuous time, they can handle arbitrarily spaced data. For the LSTM and Fully Connected models, we concatenated the times of the datapoints to the associated data and fed this into the model. For each batch of training and test data and some value of 'jitter', $j$, we create a new time series $\{t_i + j_i\}_{i=1}^{T}$, where $j_i \sim U([-j, j])$. Since our timestep for synthetic experiments is $0.5$s we try values of $j$ of $0.1$s and $0.25$s. We then generate the batch of trajectories by integrating our systems to the new timesteps.

## A.11 Tables of Results

## A.12 Additional Control Results

| System | Lorenz | | Ballistic | |
|---|---|---|---|---|
| Jitter Level (s) | 0.1 | 0.25 | 0.1 | 0.25 |
| FC | 0.104 | 0.109 | 0.106 | 0.125 |
| FC NODE | 0.008 | 0.012 | 0.075 | 0.077 |
| Full NDS | $-0.010$ | $\mathbf{-0.066}$ | $\mathbf{-0.012}$ | $\mathbf{-0.096}$ |
| Partial NDS | $\mathbf{-0.011}$ | $-0.012$ | 0.024 | 0.016 |
| LSTM | 0.032 | 0.031 | 0.130 | 0.132 |
| SR | 2.69 | 2.17 | 0.239 | 0.342 |
| GBO | $-0.002$ | 0.058 | 0.002 | 0.013 |

Table 4: The effect of jitter on a given algorithm for a given level of jitter as computed in A.10. The effect is calculated by dividing the performance with jitter by the mean performance without and subtracting 1.

| Model | $L2$ Error on the Fusion Test Set |
|---|---|
| FC | $1.942 \times 10^{11}$ |
| FC NODE | $8.288 \times 10^{10}$ |
| NDS with Approximate Dynamics | $\mathbf{4.837 \times 10^{10}}$ |
| NDS0 | $8.944 \times 10^{10}$ |
| LSTM | $2.527 \times 10^{11}$ |
| SR | $2.546 \times 10^{12}$ |
| GBO | $6.452 \times 10^{11}$ |

Table 5: The performance of our comparison models on the nuclear fusion problem, as discussed in Section 5.2.

| | MSE of Model | | Performance of MPC | |
|---|---|---|---|---|
| # of Training Examples | 100K | 25K | 100K | 25K |
| FC | $\mathbf{0.016 \pm 0.004}$ | $0.022 \pm 0.006$ | $\mathbf{86 \pm 4}$ | $76 \pm 4$ |
| FC NODE | $\mathbf{0.015 \pm 0.008}$ | $\mathbf{0.019 \pm 0.007}$ | $\mathbf{87 \pm 4}$ | $\mathbf{78 \pm 4}$ |
| LSTM | $0.081 \pm 0.023$ | $0.092 \pm 0.025$ | $33 \pm 6$ | $28 \pm 5$ |
| Full NDS | $\mathbf{0.015 \pm 0.007}$ | $\mathbf{0.017 \pm 0.002}$ | $\mathbf{86 \pm 3}$ | $\mathbf{82 \pm 5}$ |
| Partial NDS | $\mathbf{0.016 \pm 0.011}$ | $\mathbf{0.017 \pm 0.005}$ | $79 \pm 6$ | $78 \pm 5$ |
| NDS0 | $\mathbf{0.013 \pm 0.009}$ | $\mathbf{0.017 \pm 0.008}$ | $76 \pm 5$ | $74 \pm 6$ |
| SR | $0.032 \pm 0.023$ | $0.034 \pm 0.017$ | $68 \pm 4$ | $66 \pm 3$ |

Table 6: Modeling and Control on the Cartpole system with large amounts of data.

