# OpenReview forum: "Neural Dynamical Systems: Balancing Structure and Flexibility in Physical Prediction"
_ICLR.cc/2021/Conference — Reject_

### Official Review · AnonReviewer2 · 2020-10-26
**Recommend to reject**

**Rating:** 5
**Confidence:** 4

**Review:**

### Summary
The paper combines gray box optimization with Neural ODE, for improved predictions of time series data from low-dimensional physics systems.

### Recommendation
The jury is still out on how to best combine prior knowledge about a physical system with learnable components, so this is an important research direction; and combining system identification/gray box optimization with Neural ODE is a reasonable approach. However, the results remain ambiguous, and most importantly, I feel the most relevant questions (see below) are not investigated.

The big questions in system identification methods, and what often limits their usefulness, are a) how to deal with systematic effects not properly accounted for in the model prior and b) how to avoid falling into local optima due to a highly nonlinear optimization landscape, and I feel these are not sufficiently addressed in this paper. First of all, most experiments reported in the main paper (5.1) are synthetic, generated with the exact model prior, on a simple low-dimensional system, without any systematic deviations or even noise. Yet even in this ideal setting, pure NeuralODE is just barely out of the confidence interval of NDS (e.g. fig. 4). The appendix contains some experiments with added uniform noise; the authors state that NDS is affected more strongly affected by noise, but the results are hard to parse, since only relative change is given. It's unclear if NDS is still performing best in this noise setting. These results cast doubts if this approach will be useful in more complicated settings.

Second, baselines. There should be a baseline for pure GBO (same model and setup as NDS, but turning off the residual, let's call it NDS0). I'd expect this baseline to be comparable or better than NDS at least on the synthetic data in 5.1, since the model is exact in those cases. Instead, there is a GBO baseline using a proprietary MATLAB GBO function, which performs surpringly bad at estimating parameters (fig. 5) for the synthetic data. The cited Ljung et al. doesn't actually describe the MATLAB GBO method, only the interface and GUI. So it's hard to know what the differences in optimization setup are, and whether the Neural-ODE residual somehow helps parameter estimation, or it MATLAB just performs pooly due to a different optimization setup. If it's the former, and NDS actually outperforms NDS0 on parameter estimation, a proper investigation and discussion of this would actually be an interesting finding.
Also, the baseline models seem to all have a different NN architecture, with different capacity and different activation. This makes it even harder to extract meaning from the results.
The only dataset which is not a toy example is the fusion dataset in 5.2. I'm not a domain expert on this, so I don't know how hard of a task this is, or what's a relevant benchmark method is for this dataset-- however, since this is real data, there must have be an attempt to fit this data to a model by the group operating the tokamak, and such a model fit should be compared to as a baseline. Second, I assume as a real-world dataset this must include measurement noise, and effects not explained by the model prior. Surprisingly however, this is also the experiment where NDS most strongly outperforms pure Neural ODE, and even more surprisingly smaller error bounds than the purely learned methods-- the complete opposite of 5.1., which found optimization through the model prior to cause a lot of instability *even without* noise. This should definitely be investigated more closely, as this could be due to anything from different smoothness, sequence length, luck, type of model ODE, or maybe even regularization effect from the real-world data; and the conclusions to draw for the value of this method would be drastically different depending on the finding.

Finally, another big surprise in this paper to me was that partial NDS strongly outperformed NDS. I'm not sure I follow the authors' explanation--if the model parameter estimation is problematically bad, why would NDS be better than NeuralODE? It is possible that this particular form of omission makes the energy landscape easier to optimize; maybe that's what the paper implies. Or if it's just about training stability due to the coupling of jointly learning residual and model parameters, this should be eliminated by pretraining parameter estimation. But this should be properly investigated, as local minima are a typical problem in system identification, and this result could even hint at a way to alleviate this issue.

In conclusion, I feel while this is an interesting topic, the actual learnings we can draw on the value of using model priors and GBO are limited. This is not a bad paper, but in its current form I don't think it meets the standards for a top ML conference.

### Further questions
- How long is the history T'? Do the baselines (NeuralODE, FC, LSTM) have access to the history data? If not, this alone might explain the performance difference
- Wouldn't you want to feed the output of g_phi into d_tau, so it's easier to compute corrections?
- what's the performance of NDS w/o the residual, or without g_phi (this should be identical to the NeuralODE baseline; is it?). How much history does it need to estimate parameters well? Does pre-training the parameter estimation module improve performance?

----

Update:
Thank you for answering my questions and running the additional NDS0 baseline, this clarifies a vital aspect of the paper.
I'm still concerned that in its current form, the learnings we can extract from the paper are limited.
The new baseline confirms the expected, that for low-dimensional synthetic tasks where the ground truth model is known, pure sysid is a perfectly fine strategy. This is not surprising, and a good sanity check to perform. Unfortunately, this is also the largest part of the experimental results, and I'm not sure we can learn that much more beyond this.
Results on 5.2 do look promising-- neither NODE nor NDS0 perform as well as NDS, and the new baseline has strengthened this result. But as the only data point demonstrating an advantage of the method, with little analysis to why and when it does well this still feels a bit limited.
I still think this line of research is very valuable, and I encourage the authors to study the properties of this approach further (e.g. noise, missing systematic terms, investigate why partial NDS performs so well in some tasks), and investigate other non-trivial domains where NDS or variants can shine.

I've raised my score to 5.

---

> ### Author Response · Authors · 2020-11-16
> **Response to Reviewer 2**
>
> Thank you for your review! We appreciate the close reading and feedback. We hope to take it into account in order to make this paper the best it can be.
>
> First, though we agree that many of our experiments are on low-dimensional identified synthetic systems, we disagree that the performance of our algorithm is close to that of the plain Neural ODE, particularly in settings where the data are limited (the eventual closeness in Figure 4 is only in the limit of many datapoints seen). See Table 2 for the full data on these systems when trained to convergence on a limited number of datapoints, where the NDS models greatly outperform the alternatives. We will do a better job clarifying the distinction in the small-data settings in the updated version.
>
> We originally plotted the absolute performances of our model under noise, but scaling issues make the effect of noise much harder to see. The NDS models still have better performance on an absolute basis than the competitors in that setting, as we note in Section A.9.
>
> On baselines, we agree that the MATLAB version of GBO performs substantially badly on our problems, though we tried a wide variety of options and approaches from their toolbox. In trying the NDS0 variant you suggested, we find that it has similar performance on the well-identified systems as the NDS model but with substantially higher variance. On the real data, it performs substantially worse, but still better than a hardcoded model we’ll discuss in the next paragraph. The variety of baselines chosen in our setup are due to a bit of search over parameter space for each class of network. We chose different activations for the ODE-based models as on the recommendation of the Neural ODE authors the Softplus activation is better than ReLU for those models. We’ll clarify these choices in the updated version of the paper.
>
> The group operating the tokamak has worked with simple physical models such as the approximate one here for control purposes in the past [1]. We also strongly outperform this approximate model with the nominal values typically used by physicists, and will include that in our updated version of this figure. We also point out that the fusion experiments don’t include error bounds--as we only have one real dataset, it would only be feasible to train the model with different initializations rather than on different datasets. So we’re not sure what the error bounds you’re mentioning wrt the fusion experiment are.
>
> We agree that it is interesting to consider how valuable the type, quantity, and smoothness of system data and prior knowledge affect the paper. We intend to add substantial discussion of these factors in our revision.
>
> We disagree with the assessment that the partial NDS strongly outperforms the full NDS. If we reference Tables 1, 2, and 6, we see that on the Ballistic and Lorenz systems, these two algorithms perform similarly in general, which we see as showing that even the partial prior knowledge substantially improves the initialization and makes the learning problem more simple. On the cartpole experiments, the slightly improved model performance of the full NDS results in a larger gap in the performance in MPC, with the full NDS outperforming by more in control, which may be expected due to compounding model errors. We are working on additional ablations to better understand this by changing which equations are held out of the partial prior knowledge as well as the aforementioned NDS0 test, which we hope will shed more light on this question.
>
> To your specific questions, T’ varies but is 32 timesteps for the modeling experiments and 8 for the control experiments. The baselines also have access to this information. We tried the connection you mentioned between g_phi and d_tau and didn’t see an improvement in performance, so removed the connection for simplicity.  We’re currently working to answer your last question and hope to do so by the end of the discussion period. Thanks once again for your questions and comments. They’ve certainly made the paper better!
>
> [1] Feedback control of stored energy and rotation with variable beam energy and perveance on DIII-D; Boyer et al, 2019

---

### Official Review · AnonReviewer3 · 2020-10-28
**Promising contributions, but their presentation needs improvement**

**Rating:** 5
**Confidence:** 4

**Review:**

The paper proposes a neural-network architecture for modeling dynamical systems that incorporates prior domain knowledge of the system's dynamics. More specifically, the main contributions are the mechanisms for incorporating such knowledge, in terms of fully or partially known structure (differential equations) of the system, which in turn positively affects the modeling performance. The results from the experimental evaluation (on 3 synthetic and one real-world experiments), in general, show that the proposed Neural Dynamical Systems (NDS), and in particular the ones trained with partial prior knowledge, have better performance than several standard benchmarks (such as NeuralODEs, LSTMs, Sparse Regression etc.).

The paper has a very strong introduction and the authors provide a good (and quite lengthy) overview of the related work. This, however, given the page-limit, has a severe consequence on the latter parts of the paper, that are poorly structured, inconsistent and hard to follow. In particular, the experimental design as well as the presentation and the discussion of the findings need to be improved.

Comments:
- Some (rather important) details of the method are missing, such as a deeper discussion and motivation related to the architecture design. In particular, what is the role of the context encoder and how does it affects the performance of the model? How does the time-span in the history encoder affects the overall performance (also why x and u in the history encoder are of different lengths)?
- What is the difference between the Full-NDS and Approx.-NDS, with respect to g(.)?  If there is none, why are these considered different? Can you clarify why x1 and u1 in the historical encoder within Approx.-NDS are of length T',  but in the other u is of length T?
- It is unclear why the reported MSE losses, in general, are so high? Are these summed (and not averaged) for each task? Even the best performing models, when using all the training examples (Fig2), have errors with magnitudes ~10^3 (Ballistic task) and ~10-10^2 (Lorenz task), which are unusually high. What is the interpretation of this? Maybe a plot of the predicted trajectories vs. the ground truth may help.
- In Fig2, the performances of the Sparse Regression and GBO are constant w.r.t the number of training samples. After checking the appendices, their performance on the smaller sample size is reported as n/a. Does this mean that they were only trained on the complete data, or something else is happening. Moreover, Sparse Regression in particular, in principle is able to model a Lorenz system quite accurately (this is also reported in the original paper Brunton et. al 2015/2016). However here it seems is orders-of-magnitude worse than the rest- how was it parameterized and did you investigated what is the output?
- The results show that Partial-NDS seem to perform well overall, therefore the details for their parameterization need to be placed in the main part, not the appendix. Nevertheless, while the authors state that they are a partial ablation of Full-NDS, given their performance there is a trade-of between the amount of partial knowledge and the overall performance. Therefore it would be useful to study this amount of prior knowledge given at input. For instance, how will other parts of the equation structure affect their performance (eg. in the Lorenz tasks if you provide x and z but not y) or what is the least amount that is needed which will still lead to good performance with small amounts of data. Also, can parts of these knowledge be approximations etc.
- The noise/irregular spacing experiments are mentioned in the beginning of Sec 5.1. Besides the figures given in A9, these findings are never properly discussed nor summarized in the main part (except "NDS does well").
- The "Sample Complexity and Overall Accuracy" segment seems out of place and a bit confusing. The paragraph discusses the Fusion experiment but Figure4 (Lorenz) is referenced. Also can you clarify the meaning of "three points on the spectrum of added structure" and what are "our system identification models"?
- Adding a summary of the main contributions w.r.t. the results would be beneficial

Other comments:
- SINDy (Brunton et. al 2015/2016) doesn't perform Sparse Symbolic Regression - but Sparse Regression.
- Consider moving the PartialNDS set-up in the main part.
- How is the Partial-NDS parameterized in the Cartpole experiment?
- In Eq(9) and (10) why is  x_dot = x_dot (and same for the other 3)?
- Last paragraph references Table 5.3. and Table A.12  which don't exist.

-------update-----
Thank you for your response. The discussion and the revised manuscript clarified some of my concerns regarding your work. I appreciate that the authors will focus on the parameterization of PartialNDS and the effect of the amount of employed prior-knowledge. However, my concerns regarding the overall performance, reported as very high errors overall, still remain. This might be due to how the experiments are designed, how the results reported or something else - but nevertheless it needs more attention and further investigation.

---

> ### Author Response · Authors · 2020-11-16
> **Response to Reviewer 3**
>
> Hi, thanks for your review! We appreciate your knowledge and thoroughness in going over our work. We hope that these comments along with subsequent changes to the paper will address your concerns with the submission.
>
> On the design of our architecture: the context encoder is necessary for residual prediction, as there needs to be a network component in the spirit of the original neural ODE, which takes as input the current state and contributes to the derivatives calculated at the current timestep. With this constraint, something looking like a context encoder is necessary to make the appropriate connection. We also added an ablation to investigate this precise point with NDS0.
>
> The history length is something we handle differently across our modeling (longer horizon) and prediction (shorter horizon) experiments and they both work well. We’re working to include more thorough data on the effect of varying this in the appendix. The reason the control and history inputs are different lengths is that we assume access to future controls for MPC purposes. We added a note to that point.
>
> The full NDS and approximate NDS are different only in the assumed accuracy of the prior knowledge model. The input for approx-NDS is a typo and we’ll fix it in the new version.
>
> We agree that the performance of the sparse regression algorithm is a bit concerning. After spending substantial time with the code provided alongside that paper, we believe the issue is that that paper uses finite differencing to compute derivatives in order to accomplish the regression for the differential equation. As we are using a timestep of 0.5 as opposed to the 0.001 used in that paper, the estimates of the derivatives computed are not good and the algorithm does not perform well.
>
> We also agree that it is interesting that in certain cases the partial NDS outperforms the full NDS. We’re currently working on exploring other settings of prior knowledge as it would be good to include more information about this in the paper.
>
> We have included a substantially larger discussion on noise, sampling intervals, and parameter identification in the main body, so as to make sure these aspects of the problem do not get overlooked. We see these models as existing on a spectrum between a feedforward neural network which has no understanding of time, a neural ODE which has an understanding of time but not the system, and the fusion, NDS, which has an approximate understanding of the system as well. We understand that this could be confusing and will clear up the writing on this and our additional system ID methods.
>
> We hope that by clearing up the writing and adding a few experiments to clear up remaining questions about the method, the strength of our contribution can be more clearly seen as well as its weaknesses. Thank you.

---

> > ### Comment · AnonReviewer3 · 2020-11-24
> > **response**
> >
> > Thank you for your response. The discussion and the revised manuscript clarified some of my concerns regarding your work.  I appreciate that the authors will focus on the parameterization of PartialNDS and the effect of the amount of employed prior-knowledge. However, my concerns regarding the overall performance, reported as very high errors overall, still remain. This might be due to how the experiments are designed, how the results reported or something else - but nevertheless it needs more attention and further investigation.

---

### Official Review · AnonReviewer4 · 2020-10-28
**Promising method for flexible physical predictions**

**Rating:** 8
**Confidence:** 3

**Review:**

This paper is well written and introduces a novel method to learn dynamical models, incorporating prior knowledge in the form of systems of ODE.

The Neural Dynamic Systems method is described in sufficient details and multiple variations are given for the handling of systems where only partial or approximate knowledge is attainable.
The experiments explore three different applications of the NDS method introduced in the paper, to a simple synthetic and noiseless physical system, a simplified fusion system where the system dynamics are approximate, and to a modified Cartpole control problem.

The experiments show promising results, and it seems likely that the machinery developed in this paper will find impactful applications in the natural sciences and in model-based RL.
I would suggest exploring alternative RL models than the Cartpole problem, such as a robotics application, where the impact of an NDS approach might lead to more interesting results.

In the related work section, I recommend adding citations to arxiv:1909.05862 and arxiv:1909.12790, which explore very different graph-based methods to tackle a similar issue of predicting the dynamics of physical systems.

All in all, it is an interesting contribution to the literature of physical predictions, and I recommend it for acceptance.

---

> ### Author Response · Authors · 2020-11-16
> **Response to Reviewer 4**
>
> Thank you for your review! We appreciate your positive feedback on the algorithm and evaluation. We agree that the results are promising and are actively exploring applications in both the natural sciences and model-based RL as you mentioned. We agree that it would be of great interest to explore these applications both on larger-scale control tasks such as HalfCheetah and then on to real control systems such as robots. We read over your suggestions and agree they are relevant, and will add them to an updated version of the paper. Thank you!

---

### Official Review · AnonReviewer1 · 2020-10-29
**interesting application, novelty of the contribution under question**

**Rating:** 4
**Confidence:** 4

**Review:**

This work proposes an hybrid framework where the dynamical system inferred in Neural ODE (NODE) is parameterised by two separated components: a component implementing known dynamics provided by a given ODE, and a free component parameterised by a neural network. The rationale behind this approach is to exploit known properties of the data as opposed to a fully data-driven approach.
Practically, the framework is simply obtained by modifying the neural dynamics of NODE to account for transformations parameterised by the given ODE, and by letting the network take care of the dimensions for which the dynamics are unknown.

The experiments are carried out on synthetic data generated from the Lorenz system, and on data representing a simplified fusion system and control experiment. The results show improved predictions whenever the dynamics are introduced in the NODE integrator, either in full or partial form, as compared to the full non-parametric implementation through neural networks.

The work is interesting and provides a convincing argument for the usability of NODE in settings different from the one proposed in the original paper. The idea of hybrid parameterisation of the dynamics is also interesting and proven to be useful in the experimental setup. However, to my opinion the contribution of this work is quite incremental. The proposed method is still a special case of NODE, obtained by using parameterised dynamics. In this sense, standard applications of NODE (with its standard implementation of ODESolve),  can be used for this purpose, by simply parameterising the latent dynamics with the functional implementing the ODE of interest.
To my understanding, the contribution of the work is beyond the methodological framework, and resides in the application on fusion systems. However, I feel that the work should provide a more thorough evaluation of the dynamics, and attempt the implementation of more complex systems.

---

> ### Author Response · Authors · 2020-11-16
> **Response to Reviewer 1**
>
> Hi, thank you for your review. We appreciate the comments and feedback. Our primary response is that we give a much more “plug-and-play” understanding of how to incorporate dynamics for a new system than the general notion of simply parameterizing the latent dynamics with the functional representing the ODE of interest, which comes with substantial attendant complexity and a large number of design parameters in its implementation.
>
> We agree in general with your characterization of the method, though we also compare to methods from the system identification literature which are primarily focused on the structure introduced in prior knowledge and don’t include the neural network component. We agree that the fusion contribution is substantial. Reduced models such as these are useful in settings where global control decisions need to be made in a quick-and-dirty manner and improving the accuracy of those models without throwing away the original simplified equations can hopefully contribute to fusion control. The physics community has used simple physical models such as ours for control purposes in past work. NDS outperforms those models with the nominal values chosen by physicists for prediction purposes and we will include that data in an update to the paper.
>
> However, we greatly disagree with the assessment that this contribution is incremental. Substantial prior work [1][2] uses the unconstrained neural ODE for dynamics modeling, but doesn’t use the knowledge that we may have about the system being modeled. In domains where data is plentiful, this may be sufficient. But when there is limited data or a clear understanding of the dynamics, it seems clear that it would be good to leverage this knowledge in prediction.
>
> However, for a researcher focused on a particular dynamics problem to figure out the appropriate way to leverage the prior knowledge within a given deep learning modeling framework by including an appropriate functional in a Neural ODE would require them to go far afield, especially when the effects of choices of architectures hadn’t been explored, the benefits as far as sample complexity goes hadn’t been made clear, and the use for control hadn’t been demonstrated. Our work greatly concretizes this vague direction into a relatively simple framework and demonstrates its value. For these reasons, we believe that this contribution is substantial and worthy of further dissemination. Thanks again.
> [1]Turbulence forecasting via Neural ODE, Portwood et al 2019
> [2]DyNODE: Neural Ordinary Differential Equations for Dynamics Modeling in Continuous Control, Alvarez et al 2020

---

### Author Response · Authors · 2020-11-16
**Comment on new uploaded version**

We’d like to thank all the reviewers for their thoughtful feedback. The general point we’d like to emphasize is that, if one has prior knowledge in the form of ODEs for the system they’re modeling and data that is not large enough for an unconstrained Neural Network or ODE to work, this is the method they should use.

We’ve uploaded an updated version of the paper with an added ablation, NDS0, which has no residual terms. We’re currently running a few more experiments to better understand the types of prior knowledge by more comprehensively evaluating the possible prior knowledge over the Lorenz system over every subset of possible equations. We’ll upload one more version once we have completed these experiments and analysis. Thanks to all.

---

> ### Comment · AnonReviewer2 · 2020-11-20
> **Update**
>
> Thank you for the responses, updates to the manuscript and running the new baseline, which was very helpful to clarify the method's properties. Still, I have remaining concerns-- I have updated my review, and raise my score from 4 to 5.

---

### Decision · Program_Chairs · 2021-01-07
**Final Decision**

**Decision:**

Reject

**Comment:**

The paper presents a framework for modeling dynamical systems by combining prior knowledge available as ODE and implemented via a differentiable solver, with statistical modules. This is a key problem consisting in complementing available partial knowledge on a physical system with information extracted from available data with agnostic statistical methods. In their framework, both the ODE parameters and the residual model parameters are learned. Experiments are performed on synthetic and on a simplified but realistic problem.

All the reviewers do agree that the topic is important and that the paper has merits and brings an interesting contribution. They highlight some weaknesses in the presentation and more importantly in the experimental assessment. Overall, this is a good paper that should still be somewhat improved for publication. The authors are encouraged to investigate further the analysis of their framework in different settings and to bring more experimental evidence.